# Inner Retinal Thinning Comparison between Branch Retinal Artery Occlusion and Primary Open-Angle Glaucoma

**DOI:** 10.3390/diagnostics13223428

**Published:** 2023-11-10

**Authors:** Gabriella De Salvo, Mohamed Oshallah, Anastasios E. Sepetis, Ramez Borbara, Giovanni William Oliverio, Alessandro Meduri, Rino Frisina, Aby Jacob

**Affiliations:** 1Ophthalmology Department, University Hospital Southampton NHS Foundation Trust, Southampton SO16 6YD, UK; mohamed.oshallah@nhs.net (M.O.); tasos.sepetis@gmail.com (A.E.S.); ramezborbara@gmail.com (R.B.); aby.jacob@uhs.nhs.uk (A.J.); 2Biomedical, Dental and Morphological and Functional Images Sciences Department, University of Messina, 98122 Messina, Italy; gioliverio@unime.it (G.W.O.); ameduri@unime.it (A.M.); 3Ophthalmology Department, University of Padova, 35128 Padova, Italy; frisinarino@gmail.com

**Keywords:** glaucoma module premium edition, hemispheric asymmetry, inner nuclear layer, inner plexiform layer, retinal artery occlusion

## Abstract

***Purpose:*** to assess the tomographic retinal layers’ thickness in eyes affected by branch retinal artery occlusion (BRAO) and to compare it to those of patients affected by primary open angle glaucoma (POAG). ***Methods:*** retrospective review of 27 patients; 16 with BRAO (16 eyes) and 11 with POAG (20 eyes) were identified among those who received SD-OCT scans, including analysis of macular retinal nerve fiber layer (mRNFL), ganglion cell layer (GCL), inner plexiform layer (IPL), inner nuclear layer (INL), neuroretinal rim (NRR), circumpapillary RNFL at 3.5 mm and hemisphere asymmetry (HA). ***Results*:** the total IPL and INL thinning difference between the two groups was statistically significant (*p* = 0.0067 and *p* < 0.0001, respectively). The HA difference for the total macular thinning, mRNFL, GCL, IPL and INL (*p* < 0.0001) was also statistically significant. The analysis of the average total retinal thinning, total mRNFL and GCL thinning showed no statistically significant difference between the two groups. ***Conclusions:*** unilateral inner retinal thinning may represent a sign of temporal BRAO, particularly for INL thinning and HA difference over 17µm in total retinal layer thinning. This information is particularly useful in the diagnosis of previous, undiagnosed BRAO and may help prevent further retinal arterial occlusion and possible cerebrovascular incidents.

## 1. Introduction

Retinal artery occlusion (RAO) is commonly caused by an embolism of the retinal artery which is usually visible in 65% of patients at presentation [1]. RAO may cause permanent loss of vision due to ischemic retinal tissue and, if affecting a branch, may result in circumscribed visual field loss [2].

Acute Branch RAO (BRAO) causes clinically detectable intracellular retinal oedema corresponding to the territory of the occluded branch and/or visualization of retinal artery emboli. Following the acute phase, the retinal oedema resolves in about 3 months and the retina could look unremarkable, especially if emboli are absent. Even a fundus fluorescein angiogram (FFA) in the established stage may show essentially normal flow due to partial recanalization of the occluded vessel [1,3]. Spectral domain optical coherence tomography (SD-OCT) could be helpful in revealing thinning of the inner retinal layers in the region where the retinal oedema was localized. These inner retinal changes, together with the resultant visual field defect, can resemble primary open-angle glaucoma (POAG) features, particularly when patients are asymptomatic and there are no evident signs of BRAO [4,5,6].

In this study, we aim to evaluate the tomographic peripapillary and macular structural changes of the inner retinal layers caused by temporal BRAO and to compare them with those caused by POAG, since both conditions affect the inner retina.

## 2. Methods

A retrospective observational cohort clinical study was set up in the ophthalmology department, University Hospital Southampton, United Kingdom. Electronic medical records (Medisoft^®^ and eDocs^®^) available from a medical retina clinic covering the period between January 2016 and April 2022 were reviewed.

The study protocol was approved by the local institutional review board (protocol number SEV/0341; date of approve 6 June 2021) and followed the tenets of the Helsinki declaration.

Two groups were selected in this study: patients with a diagnosis of BRAO and a cohort of POAG patients who underwent disc and macula scans with the Spectralis (Heidelberg Engineering, Heidelberg, Germany) spectral domain optical coherence tomography (SD-OCT), obtained with glaucoma module premium edition (GMPE) software 4 to 6 weeks from initial presentation. The fellow eye of patients included in the BRAO group was used as a control group.

### 2.1. Inclusion Criteria

The inclusion criteria in the BRAO cohort were confirmed temporal BRAO presented acutely in our emergency department and showing retinal intracellular oedema in the territory of the occluded artery.

Meanwhile, the POAG cohort included age-matched individuals with confirmed POAG status, including clinical findings consistent with glaucomatous optic neuropathy and glaucomatous visual field defects.

### 2.2. Exclusion Criteria

Patients with incomplete medical records (lack of follow-up, transfer of care to another hospital, lack of SD-OCT scans), combined pathology, any other optic nerve pathology and/or macular pathology (i.e., age-related macular degeneration, diabetic retinopathy, myopia > 6 diopters, myopic retinal degeneration) were excluded. 

Patients or the public were not involved in the design, or conduct, or reporting or dissemination plans of our research.

### 2.3. Spectralis GMPE

SD-OCT Spectralis allows segmentation of the retinal layers and follow-up features to identify the same portion of the retina and disc with a double tracking system: a tool helpful in identifying localized inner retinal thinning in the territory of the occluded retinal artery [4]. 

In greater detail, GMPE combines the Anatomic Positioning System (APS) with individualized scan patterns to assess the optic nerve head and the inner retinal layers in the macula. The APS detects the unique anatomic shape for the posterior segment for each eye and uses these features to acquire reproducible accurate scans of the fundus using two landmarks: the center of the fovea and the center of Bruch’s membrane opening (BMO) (Figure 1) [7].

It then ensures that the scans are oriented around the fovea-to-Bruch’s membrane opening center (FoBMOC) axis that is unique for every patient. It thereby provides consistent, accurate placement of subsequent scans and sectors for data analysis.

BMO-centered 24 radial scans are acquired to provide measurement of the ONH.

Three circles’ scans centered on the BMO-demarcated optic nerve are acquired to provide RNFL thickness.

The GMPE software then compares the resulting scans against a reference database of healthy eyes to detect abnormal changes [8].

The posterior pole asymmetry analysis (PPAA), another feature of the software, combines mapping of the posterior pole retinal thickness with asymmetry analysis between eyes and between hemispheres of each eye. It also allows a layer-by-layer and sector-by-sector assessment of the macula. It provides thickness of total macular layers, macular retinal nerve fiber layer (mRNFL), ganglion cell layer (GCL) (Figure 2), inner plexiform layer (IPL), inner nuclear layer (INL) (Figure 3), outer plexiform layer (OPL), outer nuclear layer and retinal pigmented epithelium at all sectors which were defined by the Early Treatment Diabetic Retinopathy Study scheme [8].

### 2.4. Data Collection

Data collected included best documented visual acuity (VA) using ETDRS protocol, thickness of inner retinal layers: mRNFL, GCL, inner IPL, INL and neuro-retinal rim (NRR), circumpapillary RNFL at 3.5 mm and hemisphere asymmetry (HA) among these layers using the GMPE software and PPAA. Scans with incomplete or missing data were excluded.

### 2.5. Statistical Analysis

Statistical analysis was performed using GraphPad Prism Version 9.1.0. Comparisons between the BRAO and the control groups were performed using Wilcoxon matched-pairs signed-rank tests. Comparisons between POAG and the control and BRAO group were performed using the Mann–Whitney test, *p* < 0.05 was considered statistically significant. Diagnostic accuracy and prognostic values of all the parameters that showed statistical significance when comparing POAG to BRAO were assessed using a ROC analysis and examining the area under the curve (AUC). We used the Holm–Sidak method for correction for multiple comparisons and the adjusted *p* values are presented. 

## 3. Results

### 3.1. BRAO Cohort

A total of 40 patients and 41 eyes with a confirmed diagnosis of BRAO were identified. Among the 41 eyes, 10 were excluded due to missing Spectralis GMPE data, 1 with bilateral BRAO, 1 with paramacular acute middle maculopathy (PAMM), 3 with a combined vein occlusion and 9 with a small macular artery occlusion. This left 17 eyes of 17 patients (nine males and eight females) that met the criteria to be included in our temporal BRAO cohort. Mean age at diagnosis of BRAO was 72 ± 11.5 years (49–88). Mean best documented VA in the affected eyes was 0.30 ± 0.33 logMAR (0.1–1) and 0.22 logMAR (0–0.6) on the fellow eyes. Twelve patients had involvement in the right eye, while five patients had involvement in the left eye. The mean intraocular pressure was 15.1 ± 4.4 mmHg.

Table 1 shows the macular inner retinal layer values. All layers evaluated were significantly thinner in the BRAO eyes compared to the controls (Table 1). Additionally, hemisphere asymmetry was significantly greater in all the layers of the BRAO group compared to the control (Table 2). Visual field defects were documented in 10 patients, revealing an altitudinal defect, corresponding to the retinal territory of the ischemic area.

Five eyes presented an inferior defect, four eyes a superior defect and one eye a paracentral superior defect.

### 3.2. POAG Cohort

In the POAG cohort, 20 eyes of 11 patients (four males and five females) met our inclusion criteria. Mean age (on the day of the first GMPE scan) was 74.82 ± 9.9 years (59–91). Mean best-documented VA was 0.27 ± 0.41 logMAR (0–1.7). Nine patients had involvement in both eyes, while one patient had involvement in the right eye and another patient had involvement in the left eye. The mean IOP was 15.8 ± 4.9 mmHg.

Stratifying the macular inner retinal layers RNFL, GCL and INL are significantly thinner compared to the control eyes (*p* = 0.0008, *p* = 0.04 and *p* = 0.04, respectively), whereas there were no statistically differences in the IPL and all layers total (Table 1). No statistically significant differences were observed in hemisphere asymmetry between the POAG and the control group.

### 3.3. Comparison between POAG and BRAO Cohort

No statistically significant difference in the age of the patients was found between the two groups (*p* = 0.53).

No significant difference was found when comparing the average total retinal thickness, mRNFL and GCL (Table 1). However, statistically significant differences appeared when comparing IPL thickness (*p* = 0.0067) and INL thickness (*p* < 0.0001), as well as HA of total retinal thickness, mRNFL GCL, IPL and INL (*p* < 0.00001, respectively) between the two groups (Table 2). All statistically significant values were assessed for their diagnostic ability [9]. Table 3 shows that IPL thickness and HA of mRNFL and IPL are moderately accurate in diagnosing BRAO with AUC 0.7–0.9. INL thickness and HA of total thickness, GCL, INL is highly accurate in discriminating BRAO from glaucoma with AUC 0.93.

As for the optic nerve OCT scans (Table 4), no statistically significant difference was found in BMO size between the two groups. The NRR and the circumpapillary RNFL were thinner in the POAG group (*p* < 0.0001 and *p* < 0.0008, respectively).

In our two-study cohort groups, NRR thinning was shown to be highly accurate in discriminating BRAO from POAG (AUC 0.90) and RNFL was moderately accurate (AUC 0.82).

## 4. Discussion

The diagnosis of BRAO is based on medical history and clinical examination findings. It is usually straightforward in the acute stage where the distal capillary bed becomes non-perfused, leading to inner retinal ischemia and loss of function of the affected inner retina. Central, paracentral, or altitudinal visual field scotomas may occur following the hypoxic event. Corresponding microaneurysmal dilatation of the neighboring capillary beds tends to leak, causing secondary stagnation of axoplasmic flow and formation of cotton wool spots in the acute stage (acute phase intraretinal oedema) [10].

Delayed or late presentation of BRAO, after the acute phase findings have resolved, may result in a relatively normal-looking fundus. There could be subtle signs making the diagnosis challenging, especially in cases where the BRAO involves a vessel in the temporal arcades, causing visual field changes similar to POAG [3,11,12].

In a longitudinal study, Kim et al. described an increased macular thickness in the occluded areas in the acute phase of BRAO followed by gradual thinning during the following months compared with the initial visit [13]. 

These macular and RNFL thickness changes were also described in patients with CRAO, showing a progression of RNFL atrophy over time, and subclinical changes in the inner retinal layer even in the fellow eye [14,15].

Furthermore, Ghazi et al. compared the macular OCT findings of 17 patients with CRAO and BRAO with 32 patients with non-acute optic neuropathy. They identified three differentiating features in the arterial occlusion group; complete inner retinal atrophy with loss of the normal stratification of the inner retinal layers, loss of the normal foveal depression and marked thinning of the involved retina up to the level of the foveal depression. These specific features were absent in the non-acute optic neuropathy patients [16]. 

As the retinal arterial circulation supplies the inner retina up to the OPL, these layers are affected in BRAO and become atrophic over time. In contrast, the outer retina remains unaffected as it receives its blood supply from the choroidal circulation [17,18].

In POAG, retinal changes are primarily confined to the NRR, RNFL and GCL [19,20].

Zangalli et al., in a cross-sectional study on patients with unilateral POAG, found significant thinning of the macular RNFL, GCC and GC-IPL layers. This was particularly significant when evaluated in vertical scans [21]. 

Moreover, Martucci et al. compared 48 eyes with glaucoma to 35 eyes of healthy subjects using the GMPE module. They demonstrated significant difference in thickness in the GCL and RNFL layers between the glaucoma and control groups [22].

In both pathologies (POAG and BRAO), there is loss of the RNFL and GCL. Our study demonstrates a reliable method to identify previous temporal BRAO which may have been undetected. To validate this data, we compared SD-OCT GMPE figures extracted from age-matched groups of patients with confirmed temporal BRAO and POAG. GMPE scans were obtained 4 to 6 weeks following the acute presentation of temporal BRAO.

Our results show a statistically significant difference in the total INL between the two cohorts. Consistent with our findings, Sullivan-Mee et al. demonstrated that occult BRAO could be differentiated from POAG using SD-OCT macular thickness parameters and intra-eye macular thickness asymmetry index [3].

Moreover, when examining the difference between the superior and inferior retina (HA map) in each group for each specific retinal layer, the HA maps for the total macular thickness, mRNFL, GCL, IPL and INL (Figure 4) showed a statistically significant difference when comparing BRAO to POAG. This method specifically highlights the hemispheric difference in retinal thickness in the aforementioned layers in the presence of previous temporal BRAO. We found that the hemispheric difference was indeed significantly higher in the BRAO group compared to the POAG group. However, the average total retinal thickness, total mRNFL, total GCL and total IPL differences between the groups were not statistically significant.

These results reflect the physiological distribution of retinal circulation. As retinal arteries supply the inner two thirds of the retina, an occlusion will cause an ischemia of the corresponding layers [4,13]. In glaucoma there is usually a generalized loss of RNFL and GCL; a focal notching in the NRR would cause a corresponding loss of the layers affected [23,24,25,26,27,28,29]. Our results confirm these findings and show a more robust correlation when evaluating the RNFL and GCL layers individually.

Our study also demonstrates that, when looking at the optic nerve analysis, the average NRR thickness (G) was significantly reduced in the POAG group compared to the BRAO group (*p* < 0.0001). This difference was less pronounced when evaluating the RNFL at the 3.5 mm zone (*p* < 0.01). 

Regarding symmetry, Sullivan-Mee et al. reported an intra-eye macular thickness asymmetry greater than 25 μm in patients with BRAO [3]. In our study, the largest hemispheric difference found in the POAG group was 17 µm, hence, we advocate that hemispheric differences above 17 µm in the total retinal layers may raise the possibility, or at least alert the clinician, of a possible temporal BRAO. 

The analysis of the BRAO cohort has proven to be more challenging when the arterial occlusion occurred at the cilioretinal artery, as this ischemic event is less likely to present hemispheric difference. However, these results should be confirmed in a larger study.

### Study Limitations

The retrospective nature, the small sample size, and the heterogeneity in the POAG group, represent some limitations of this study. Furthermore, not all enrolled patients had their visual field indexes evaluated and correlated to the SD-OCT parameters, as this exam is not routinely performed in our clinic in all the patients presenting with BRAO. Despite the limitations, the results of this study could aid in differentiating between BRAO and glaucomatous optic neuropathy. Retinal tomographic changes presumed to be of a different etiology may actually result from a previous temporal BRAO. Ensuing medical workup and laboratory investigations for embolic sources may prevent further retinal arterial occlusions and other potential vascular incidents. Conversely, patients initially diagnosed with BRAO and investigated accordingly may actually be suffering from other conditions such as POAG or normal tension glaucoma, requiring different treatment approaches.

Future larger prospective studies and electrophysiological correlations are essential to establish this method and provide more precise diagnostic cut-off values.

## Figures and Tables

**Figure 1 diagnostics-13-03428-f001:**
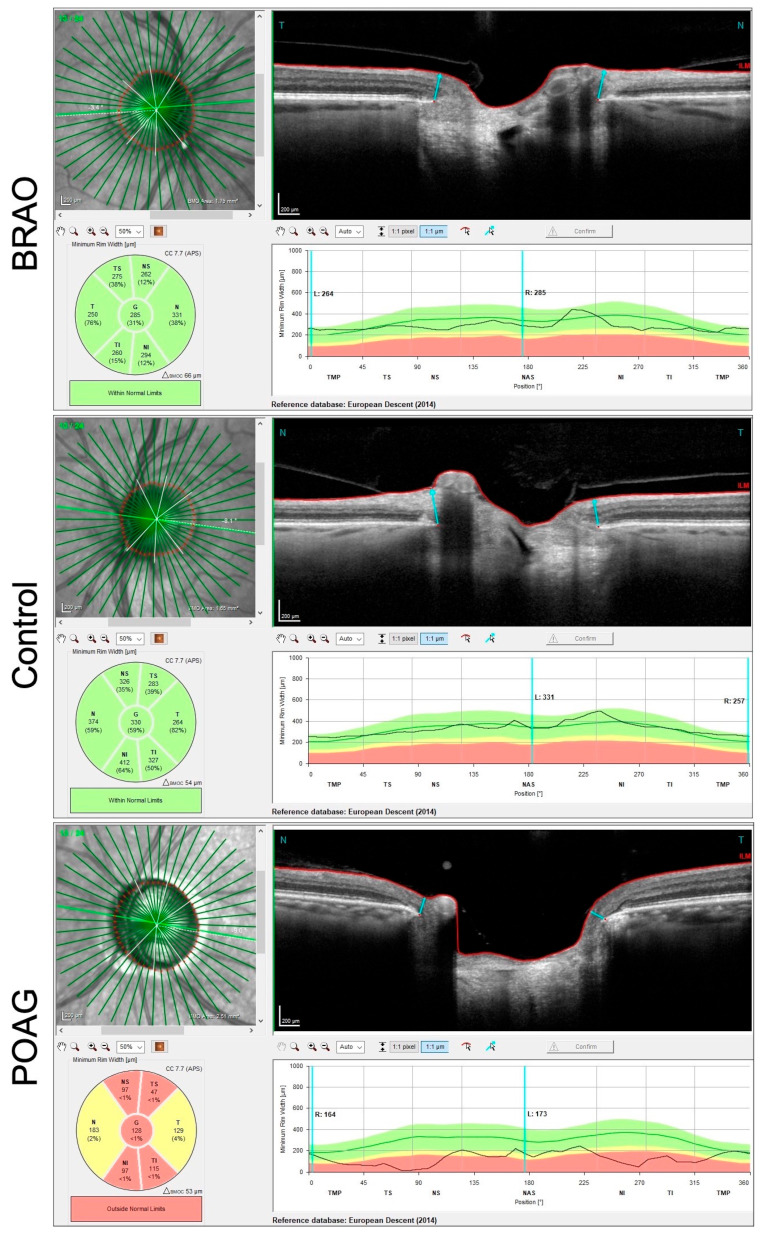
Bruch’s membrane opening (area between the two blue arrows) and the NRR map showing normal thickness in a BRAO patient and control (green colours) and thinning (yellow and red colours) in a POAG patient.

**Figure 2 diagnostics-13-03428-f002:**
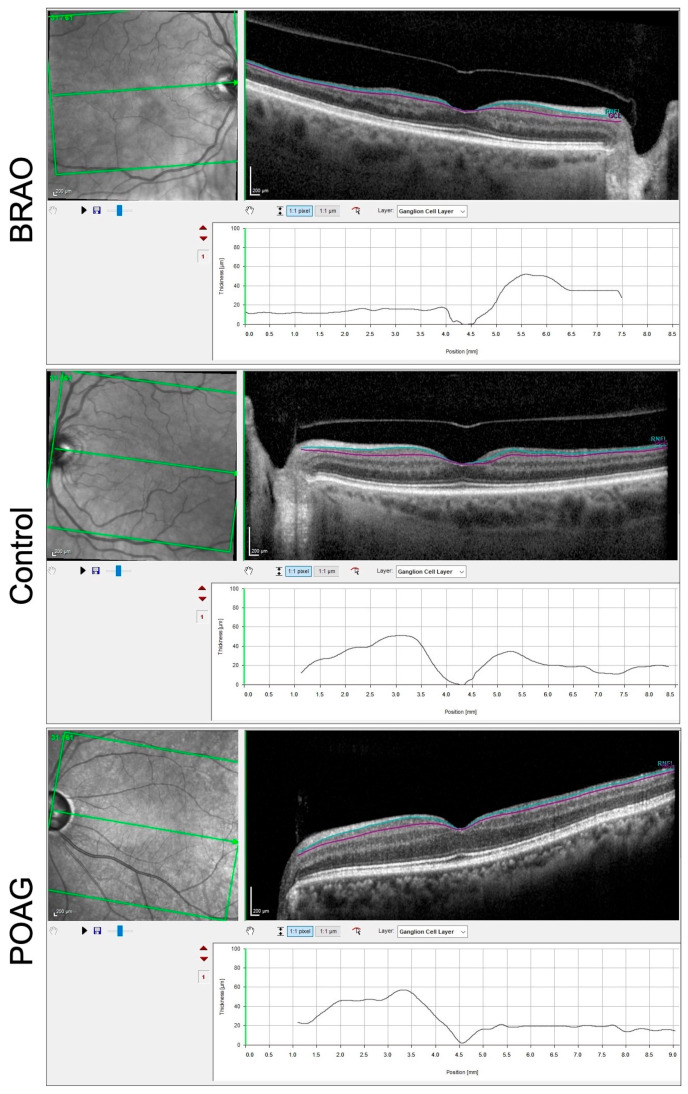
SD-OCT of a right BRAO showing thinning of GCL (area between the teal and purple lines in each of the cases shown) visible on segmentation, normal thickness of GCL in a normal patient (control) and normal thickness of GCL in a POAG patient.

**Figure 3 diagnostics-13-03428-f003:**
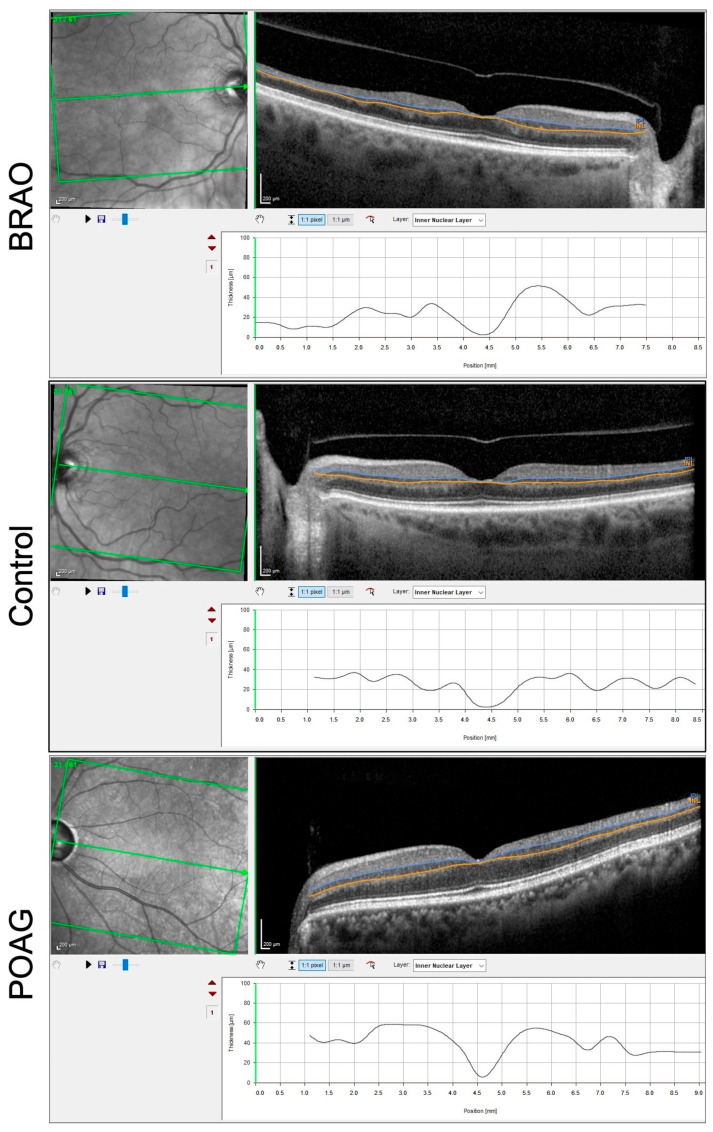
SD-OCT of a right BRAO showing thinning of INL (area between the cerulean blue and yellow lines in each of the cases shown) visible on segmentation, normal thickness of INL in a control patient and normal thickness of INL in a POAG patient.

**Figure 4 diagnostics-13-03428-f004:**
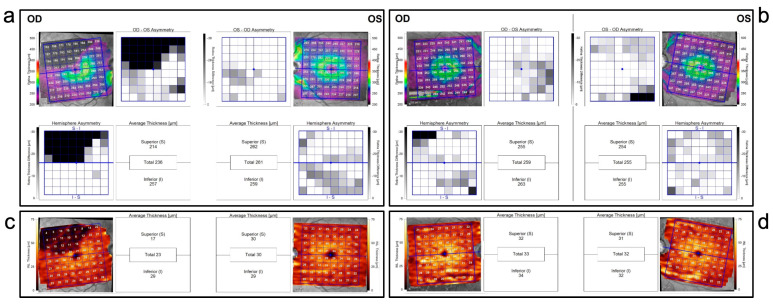
(**a**) Patient with a right (OD) supra-temporal branch retinal artery occlusion. Hemisphere asymmetry map for the total macular thickness compares the thickness of the upper and lower temporal retinal quadrants showing the difference in thickness between the two quadrants (average of 40 µm) in OD. (**c**) Posterior pole asymmetry analysis of the same patient shows significant thinning of the inner nuclear layer (INL) in the territory of the occlusion in OD. The yellow scale indicates healthy INL, while in OD the black indicates loss of INL. (**b**) Primary open angle glaucoma patient. Hemisphere asymmetry map for the total macular thickness compares the thickness of the upper and lower temporal retinal quadrants, showing no significant hemispherical asymmetry between the two quadrants. (**d**) Posterior pole asymmetry analysis of the same patient shows no significant thinning of the inner nuclear layer (INL) in each hemisphere. The yellow scale indicates healthy INL.

**Table 1 diagnostics-13-03428-t001:** Comparison of the average total retinal, macular retinal nerve fiber layer (mRNFL), ganglion cell layer (GCL), inner plexiform layer (IPL) and inner nuclear layer (INL) thickness between branch retinal artery occlusion (BRAO) group, BRAO fellow eye (control) group and primary open angle glaucoma (POAG) group. All values are represented as mean ± SD in μm. All layers are significantly thinner in the BRAO eye compared to the fellow normal eyes (Wilcoxon matched-pairs signed-rank test). In POAG RNFL, GCL and INL are significantly thinner compared to the control eyes (Mann–Whitney test). Comparing POAG with BRAO eyes, IPL and INL are significantly thinner in BRAO (Mann–Whitney test).

	Mean Values ± SD			*p* Value		
	BRAO	Control	POAG	BRAO vs. Control	POAG vs. Control	POAG vs. BRAO
**All layers total**	265 ± 20	285 ± 25	273 ± 16	0.0016	0.1703	0.2024
**mRNFL total**	32 ± 6.0	39 ± 6.3	29 ± 7.9	0.0005	0.0008	0.0927
**GCL total**	24 ± 3.9	30 ± 5.0	27 ± 5.8	0.0006	0.0478	0.1234
**IPL total**	23 ± 3.3	26 ± 4.9	25 ± 4.7	0.0027	0.2349	0.0067
**INL total**	25 ± 3.0	31 ± 3.9	32 ± 3.2	0.0039	0.0424	<0.0001

**Table 2 diagnostics-13-03428-t002:** Comparison of hemispherical asymmetry of all retinal layers, macular retinal nerve fiber layer (mRNFL), ganglion cell layer (GCL), inner plexiform layer (IPL) and inner nuclear layer (INL) between branch retinal artery occlusion (BRAO) group, BRAO fellow eye (control) group and primary open angle glaucoma (POAG) group. Values are represented as mean ± SD in μm. Hemisphere asymmetry is significantly greater in all the layers of the BRAO group compared to the control (Wilcoxon matched-pairs signed-rank test) and the POAG group (Mann–Whitney test). There is no difference between the POAG and the control group.

	Mean Values ± SD			*p* Value		
	BRAO	Control	POAG	BRAO vs. Control	POAG vs. Control	POAG vs. BRAO
**All layers-Hemisphere** **asymmetry**	38 ± 18	5.6 ± 5.5	7.4 ± 6/0	<0.0001	0.4817	<0.0001
**mRNFL-Hemisphere** **asymmetry**	15 ± 8.1	5.7 ± 3.2	5.3 ± 4.4	0.0005	0.4904	<0.0001
**GCL-Hemisphere** **asymmetry**	10 ± 4.1	1.8 ± 2.8	2.1 ± 1.6	<0.0001	0.1177	<0.0001
**IPL-Hemisphere** **asymmetry**	5.3 ± 2.7	1.2 ± 0.91	1.3 ± 1.2	0.0004	0.9434	<0.0001
**INL-Hemisphere** **asymmetry**	9.9 ± 3.9	1.1 ± 1.8	0.75 ± 0.85	<0.0001	0.8303	<0.0001

**Table 3 diagnostics-13-03428-t003:** Area under the curve analysis for inner nuclear layer thickness, neuroretinal rim (NRR) thinning, retinal nerve fiber layer (RNFL) thinning and the hemispherical difference of total retinal thickness, macular retinal nerve fiber layer (mRNFL), ganglion cell layer (GCL), inner plexiform layer (IPL) and INL for the two groups. *Test accuracy: a. 0.9  <  AUC  <  1.0, the test is highly accurate, b. 0.7  <  AUC  <  0.9, the test is moderately accurate, c. 0.5  <  AUC  <  0.7, the test is not accurate, d. AUC  =  0.5, the test is not informative.*

Parameter	AUC	Test Accuracy	*p* Value
All layers Hemisphere difference	0.9563	a	<0.0001
mRNFL Hemisphere difference	0.8844	b	<0.0001
GCL Hemisphere difference	0.9720	a	<0.0001
IPL total	0.7594	b	0.0082
IPL Hemisphere difference	0.8953	b	<0.0001
INL total	0.9672	a	<0.0001
INL Hemisphere difference	0.9781	a	<0.0001
NRR Thinning (G)	0.9000	a	<0.0001
RNFL Thinning at 3.50 mm (G)	0.8188	b	0.0012

**Table 4 diagnostics-13-03428-t004:** Comparison of branch retinal artery (BRAO) group, BRAO fellow eye (control) and primary open angle glaucoma (POAG) groups in terms of Bruch’s membrane opening (BMO) area, average value (G) of neuroretinal rim (NRR) thinning and average value (G) of retinal nerve fiber layer (RNFL), thinning at 3.50 mm circle on spectral domain optical coherence tomography (SD-OCT). All values are represented as mean ± SD in μm. NRR thinning (G) and RNFL thinning at 3.50 mm (G) are significantly reduced in BRAO compared to the control group (Wilcoxon matched-pairs signed-rank test) and even more reduced in POAG compared to the other two groups (Mann–Whitney test).

	Mean Values ± SD			*p* Value		
	BRAO	Control	POAG	BRAO vs. Control	POAG vs. Control	POAG vs. BRAO
**BMO area**	1.9 ± 0.44	1.9 ± 0.48	2 ± 0.43	0.99	0.9686	0.8689
**NRR Thinning (G)**	257 ± 45	281 ± 48	167 ± 52	0.006	<0.0001	<0.0001
**RNFL Thinning at** **3.50 mm (G)**	87 ± 14	95 ± 11	67 ± 17	0.0245	<0.0001	0.0008

## Data Availability

The datasets analyzed during the current study are available from the corresponding author on reasonable request.

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
