# Peer review of "Inner Retinal Thinning Comparison between Branch Retinal Artery Occlusion and Primary Open-Angle Glaucoma"

_diagnostics, 2023, doi:10.3390/diagnostics13223428_

Round 1
Reviewer 1 Report
Comments and Suggestions for Authors
The entitled: Inner retina thinning in branch retinal artery occlusion and primary open angle glaucoma
Major comments:
The article contains comparison of clinical information of BRAO and POAG. However, description of result is not fancy. Here are included suggestions. The BRAO patients has performed visual field looks similar to POAG? POAG is various different according to visual field damage and BRAO has various condition. This study can be arbitrary data if the authors can provide adequate information in the study. And reconstruction of discussion section is necessary.
Comments:
1. More precise correction of title enhances the study’s purpose. The reason of study should be declared by the authors. Comparison of inner retinal structure between BRAO and POAG?
2. The patients’ demographic data should be present including eye examination visual acuity and IOP, since retinal thickness is different to according to age and ethnic groups.
3. Table 1 and Table 2 changes as figures for better understanding.
4. Providing of SD-OCT scan area and retinal layer segmentation of cases of POAG and BRAO and control enhance understanding.
5. BMO area NRR image should be provide as method section as figures.
6. The quality of Figure 1 should be increased. As the reviewer mentioned, the damage of BRVO and POAG is different in posterior pole asymmetry analysis. Please check the retinal thickness study of POAG (Detection of localized retinal nerve fiber layer defects with posterior pole asymmetry analysis of spectral domain optical coherence tomography: PMID 22577076, in Investigative Ophthalmology & Visual Science).
Comments on the Quality of English LanguageDiscussion section should be re-written, since quality of English presentation is poor!
Author Response
- More precise correction of title enhances the study’s purpose. The reason of study should be declared by the authors. Comparison of inner retinal structure between BRAO and POAG?
Thank you for this advice, the title was edited according with your suggestion to: ‘Inner retinal thinning comparison between branch retinal artery occlusion and primary open-angle glaucoma’.
- The patients’ demographic data should be present including eye examination visual acuity and IOP, since retinal thickness is different to according to age and ethnic groups.
The patient’s demographics are implemented, as required, we included IOP as requested, see results section highlighted in yellow.
- Table 1 and Table 2 changes as figures for better understanding.
Thank you for your advice. We agree with your point that figures can provide immediate clarity. Nevertheless, when we attempted the figures, the graphs’ plotting was somewhat confusing, therefore we opted to keep the original version with tables, which provides a detailed description of our data.
- Providing of SD-OCT scan area and retinal layer segmentation of cases of POAG and BRAO and control enhance understanding.
- BMO area NRR image should be provide as method section as figures.
We have included a detailed description of the scan considered in the GMPE module, along with three new figures to provide a clearer representation of retinal layer segmentation and the BMO (Bruch's Membrane Opening) area NRR (Neuroretinal Rim) image as suggested.
- The quality of Figure 1 should be increased. As the reviewer mentioned, the damage of BRVO and POAG is different in posterior pole asymmetry analysis. Please check the retinal thickness study of POAG (Detection of localized retinal nerve fiber layer defects with posterior pole asymmetry analysis of spectral domain optical coherence tomography: PMID 22577076, in Investigative Ophthalmology & Visual Science).
We have included a better-quality figure. Furthermore, we discussed and included the suggested reference.
Reviewer 2 Report
Comments and Suggestions for Authors
This is a very interesting study on the retinal layers' thickness differences as assessed by the OCT between patients with BRAO and POAG. These differences could potentially assist in differentiating these two entities with similar clinical findings. The inner plexiform and the inner layer layer were the two variables more significantly different between the two groups.
The study is well performed and presented. Major limitations of the study are the respective design (inserting misclassification bias) and the small sample size of the two groups ( more differences could probably have been found with a larger sample size). Despite that, the differences found are scientifically plausible.
In all tables' headings, the authors use both the POAG and glaucoma terms. They should choose one of these, better the POAG one (if they are certain that all the cases were POAG).
Author Response
This is a very interesting study on the retinal layers' thickness differences as assessed by the OCT between patients with BRAO and POAG. These differences could potentially assist in differentiating these two entities with similar clinical findings. The inner plexiform and the inner layer layer were the two variables more significantly different between the two groups.
The study is well performed and presented. Major limitations of the study are the respective design (inserting misclassification bias) and the small sample size of the two groups ( more differences could probably have been found with a larger sample size). Despite that, the differences found are scientifically plausible.
In all tables' headings, the authors use both the POAG and glaucoma terms. They should choose one of these, better the POAG one (if they are certain that all the cases were POAG).
Many thanks for your constructive comments. We discussed the main limitations of this study and made the changes suggested, replacing 'in POAG' throughout the text as recommended.
Round 2
Reviewer 1 Report
Comments and Suggestions for Authors
The reviewer believe the quality of the paper has improved after the major revision.
Minor comments:
1. It should be clearly stated whether descriptive statistics in Tables 1 and 2 table 4 are presented as "median with IQR" or "mean with SD". current mean and total range is not popular presentation form.
2. In table 2 and 4 some numbers contain comma, which is difficult to understand. should be it period?
3. P value can be NS, significant level * or numbers, which is not necessary to be in same table (table 1, 2, 4). Generally P<0.05 is regards as cut-off.
Comments on the Quality of English LanguageDiscussion section requires English editing. Many grammar error is easy found for article.
Author Response
Many thanks for your advice. We have revised the manuscript according to your suggestions.